# From Parity to Preference-based Notions of Fairness in Classification

**Muhammad Bilal Zafar**
MPI-SWS
mzafar@mpi-sws.org

**Isabel Valera**
MPI-IS
isabel.valera@tue.mpg.de

**Manuel Gomez Rodriguez**
MPI-SWS
manuelgr@mpi-sws.org

**Krishna P. Gummadi**
MPI-SWS
gummadi@mpi-sws.org

**Adrian Weller**
University of Cambridge & Alan Turing Institute
aw665@cam.ac.uk

## Abstract

The adoption of automated, data-driven decision making in an ever expanding range of applications has raised concerns about its potential unfairness towards certain social groups. In this context, a number of recent studies have focused on defining, detecting, and removing unfairness from data-driven decision systems. However, the existing notions of fairness, based on *parity* (equality) in treatment or outcomes for different social groups, tend to be quite stringent, limiting the overall decision making accuracy. In this paper, we draw inspiration from the fair-division and envy-freeness literature in economics and game theory and propose *preference*-based notions of fairness—given the choice between various sets of decision treatments or outcomes, any group of users would collectively prefer its treatment or outcomes, regardless of the (dis)parity as compared to the other groups. Then, we introduce tractable proxies to design margin-based classifiers that satisfy these preference-based notions of fairness. Finally, we experiment with a variety of synthetic and real-world datasets and show that preference-based fairness allows for greater decision accuracy than parity-based fairness.

## 1 Introduction

As machine learning is increasingly being used to automate decision making in domains that affect human lives (*e.g.*, credit ratings, housing allocation, recidivism risk prediction), there are growing concerns about the potential for *unfairness* in such algorithmic decisions [23, 25]. A flurry of recent research on fair learning has focused on defining appropriate notions of fairness and then designing mechanisms to ensure fairness in automated decision making [12, 14, 18, 19, 20, 21, 28, 32, 33, 34].

Existing notions of fairness in the machine learning literature are largely inspired by the concept of **discrimination** in social sciences and law. These notions call for **parity** (*i.e.*, equality) in **treatment**, in **impact**, or both. To ensure parity in treatment (or treatment parity), decision making systems need to avoid using users' sensitive attribute information, *i.e.*, avoid using the membership information in socially salient groups (*e.g.*, gender, race), which are protected by anti-discrimination laws [4, 10]. As a result, the use of group-conditional decision making systems is often prohibited. To ensure parity in impact (or impact parity), decision making systems need to avoid disparity in the fraction of users belonging to different sensitive attribute groups (*e.g.*, men, women) that receive *beneficial* decision outcomes. A number of learning mechanisms have been proposed to achieve parity in treatment [24],

---

An open-source code implementation of our scheme is available at: http://fate-computing.mpi-sws.org/

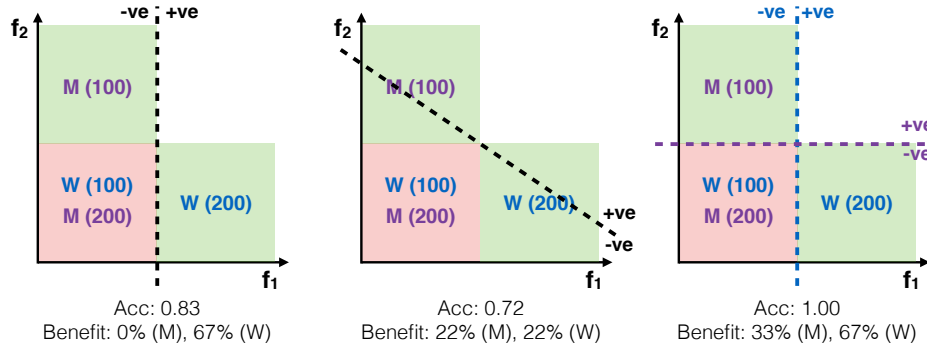

Figure 1: A fictitious decision making scenario involving two groups: men (M) and women (W). Feature $f_1$ (x-axis) is highly predictive for women whereas $f_2$ (y-axis) is highly predictive for men. Green (red) quadrants denote the positive (negative) class. Within each quadrant, the points are distributed uniformly and the numbers in parenthesis denote the number of subjects in that quadrant. The **left panel** shows the optimal classifier satisfying parity in treatment. This classifier leads to all the men getting classified as negative. The **middle panel** shows the optimal classifier satisfying parity in impact (in addition to parity in treatment). This classifier achieves impact parity by misclassifying women from positive class into negative class, and in the process, incurs a significant cost in terms of accuracy. The **right panel** shows a classifier consisting of group-conditional classifiers for men (purple) and women (blue). Both the classifiers satisfy the preferred treatment criterion since for each group, adopting the other group's classifier would lead to a smaller fraction of beneficial outcomes. Additionally, this group-conditional classifier is also a preferred impact classifier since both groups get more benefit as compared to the impact parity classifier. The overall accuracy is better than the parity classifiers.

parity in impact [7, 18, 21] or both [12, 14, 17, 20, 32, 33, 34]. However, these mechanisms pay a significant cost in terms of the accuracy (or utility) of their predictions. In fact, there exist some inherent tradeoffs (both theoretical and empirical) between achieving high prediction accuracy and satisfying treatment and / or impact parity [9, 11, 15, 22].

In this work, we introduce, formalize and evaluate new notions of fairness that are inspired by the concepts of **fair division** and **envy-freeness** in economics and game theory [5, 26, 31]. Our work is motivated by the observation that, in certain decision making scenarios, the existing parity-based fairness notions may be too stringent, precluding more accurate decisions, which may also be desired by every sensitive attribute group. To relax these parity-based notions, we introduce the concept of a user **group's preference** for being assigned one set of decision outcomes over another. Given the choice between various sets of decision outcomes, any group of users would collectively *prefer* the set that contains *the largest fraction* (or the greatest number) of beneficial decision outcomes for that group.[1] More specifically, our new preference-based notions of fairness, which we formally define in the next section, use the concept of user group's preference as follows:

— **From Parity Treatment to Preferred Treatment:** To offer preferred treatment, a decision making system should ensure that every sensitive attribute group (*e.g.*, men and women) *prefers* the set of decisions they receive over the set of decisions they would have received had they collectively presented themselves to the system as members of a different sensitive group.

The preferred treatment criterion represents a relaxation of treatment parity. That is, every decision making system that achieves treatment parity also satisfies the preferred treatment condition, which implies (in theory) that the optimal decision accuracy that can be achieved under the preferred treatment condition is at least as high as the one achieved under treatment parity. Additionally, preferred treatment allows group-conditional decision making (not allowed by treatment parity), which is necessary to achieve high decision accuracy in scenarios when the predictive power of features varies greatly between different sensitive user groups [13], as shown in Figure 1.

While preferred treatment is a looser notion of fairness than treatment parity, it retains a core fairness property embodied in treatment parity, namely, *envy-freeness at the level of user groups*. Under preferred treatment, no group of users (*e.g.*, men or women, blacks or whites) would feel that they would be collectively better off by switching their group membership (*e.g.*, gender, race). Thus,

preferred treatment decision making, despite allowing group-conditional decision making, is not vulnerable to being characterized as "reverse discrimination" against, or "affirmative action" for certain groups.

— **From Parity Impact to Preferred Impact:** To offer preferred impact, a decision making system needs to ensure that every sensitive attribute group (*e.g.*, men and women) *prefers* the set of decisions they receive over the set of decisions they would have received under the criterion of impact parity.

The preferred impact criterion represents a relaxation of impact parity. That is, every decision making system that achieves impact parity also satisfies the preferred impact condition, which implies (in theory) that the optimal decision accuracy that can be achieved under the preferred impact condition is at least as high as the one achieved under impact parity. Additionally, preferred impact allows disparity in benefits received by different groups, which may be justified in scenarios where insisting on impact parity would only lead to a reduction in the beneficial outcomes received by one or more groups, without necessarily improving them for any other group. In such scenarios, insisting on impact parity can additionally lead to a reduction in the decision accuracy, creating a case of tragedy of impact parity with a worse decision making all round, as shown in Figure 1.

While preferred impact is a looser notion of fairness compared to impact parity, by guaranteeing that every group receives *at least* as many beneficial outcomes as they would would have received under impact parity, it retains the core fairness gains in beneficial outcomes that discriminated groups would have achieved under the fairness criterion of impact parity.

Finally, we note that our preference-based fairness notions, while having many attractive properties, are not the most suitable notions of fairness in *all* scenarios. In certain cases, parity fairness may well be the eventual goal [3] and the more desirable notion.

In the remainder of this paper, we formalize our preference-based fairness notions in the context of binary classification (Section 2), propose tractable and efficient proxies to include these notions in the formulations of convex margin-based classifiers in the form of convex-concave constraints (Section 3), and show on several real world datasets that our preference-based fairness notions can provide significant gains in overall decision making accuracy as compared to parity-based fairness (Section 4).

## 2 Defining preference-based fairness for classification

In this section, we will first introduce two useful quality metrics—*utility* and *group benefit*—in the context of fairness in classification, then revisit parity-based fairness definitions in the light of these quality metrics, and finally formalize the two preference-based notions of fairness introduced in Section 1 from the perspective of the above metrics. For simplicity, we consider binary classification tasks, however, the definitions can be easily extended to m-ary classification.

**Quality metrics in fair classification.** In a fair (binary) classification task, one needs to find a mapping between the user feature vectors $\boldsymbol{x} \in \mathbb{R}^d$ and class labels $y \in \{-1, 1\}$, where $(\boldsymbol{x}, y)$ are drawn from an (unknown) distribution $f(\boldsymbol{x}, y)$. This is often achieved by finding a mapping function $\boldsymbol{\theta} : \mathbb{R}^d \to \mathbb{R}$ such that given a feature vector $\boldsymbol{x}$ with an unknown label $y$, the corresponding classifier predicts $\hat{y} = \text{sign}(\boldsymbol{\theta}(\boldsymbol{x}))$. However, this mapping function also needs to be *fair* with respect to the values of a user sensitive attribute $z \in \mathcal{Z} \subseteq \mathbb{Z}_{\geq 0}$ (*e.g.*, sex, race), which are drawn from an (unknown) distribution $f(z)$ and may be dependent of the feature vectors and class labels, *i.e.*, $f(\boldsymbol{x}, y, z) = f(\boldsymbol{x}, y|z)f(z) \neq f(\boldsymbol{x}, y)f(z)$.

Given the above problem setting, we introduce the following quality metrics, which we will use to define and compare different fairness notions:

I. **Utility ($\mathcal{U}$)**: overall profit obtained by the decision maker using the classifier. For example, in a loan approval scenario, the decision maker is the bank that gives the loan and the utility can be the overall accuracy of the classifier, *i.e.*:

$$\mathcal{U}(\boldsymbol{\theta}) = \mathbb{E}_{\boldsymbol{x},y}[\mathbb{I}\{\text{sign}(\boldsymbol{\theta}(\boldsymbol{x})) = y\}],$$

where $\mathbb{I}(\cdot)$ denotes the indicator function and the expectation is taken over the distribution $f(\boldsymbol{x}, y)$. It is in the decision maker's interest to use classifiers that maximize utility. Moreover, depending on the scenario, one can attribute different profit to true positives and true negatives— or conversely, different cost to false negatives and false positives—while computing utility. For

example, in the loan approval scenario, marking an eventual defaulter as non-defaulter may have a higher cost than marking a non-defaulter as defaulter. For simplicity, in the remainder of the paper, we will assume that the profit (cost) for true (false) positives and negatives is the same.

II. **Group benefit** ($\mathcal{B}_z$): the fraction of beneficial outcomes received by users sharing a certain value of the sensitive attribute $z$ (*e.g.*, blacks, hispanics, whites). For example, in a loan approval scenario, the beneficial outcome for a user may be receiving the loan and the group benefit for each value of $z$ can be defined as:

$$\mathcal{B}_z(\boldsymbol{\theta}) = \mathbb{E}_{\boldsymbol{x}|z}[\mathbb{I}\{\text{sign}(\boldsymbol{\theta}(\boldsymbol{x})) = 1\}],$$

where the expectation is taken over the conditional distribution $f(\boldsymbol{x}|z)$ and the bank offers a loan to a user if $\text{sign}(\boldsymbol{\theta}(\boldsymbol{x})) = 1$. Moreover, as suggested by some recent studies in fairness-aware learning [18, 22, 32], the group benefits can also be defined as the fraction of beneficial outcomes conditional on the true label of the user. For example, in a recidivism prediction scenario, the group benefits can be defined as the fraction of eventually non-offending defendants sharing a certain sensitive attribute value getting bail, that is:

$$\mathcal{B}_z(\boldsymbol{\theta}) = \mathbb{E}_{\boldsymbol{x}|z,y=1}[\mathbb{I}\{\text{sign}(\boldsymbol{\theta}(\boldsymbol{x})) = 1\}],$$

where the expectation is taken over the conditional distribution $f(\boldsymbol{x}|z, y = 1)$, $y = 1$ indicates that the defendant does not re-offend, and bail is granted if $\text{sign}(\boldsymbol{\theta}(\boldsymbol{x})) = 1$.

**Parity-based fairness.** A number of recent studies [7, 14, 18, 21, 32, 33, 34] have considered a classifier to be fair if it satisfies the impact parity criterion. That is, it ensures that the group benefits for all the sensitive attribute values are equal, *i.e.*:

$$\mathcal{B}_z(\boldsymbol{\theta}) = \mathcal{B}_{z'}(\boldsymbol{\theta}) \quad \text{for all } z, z' \in \mathcal{Z}. \tag{1}$$

In this context, different (or often same) definitions of group benefit (or beneficial outcome) have lead to different terminology, *e.g.*, disparate impact [14, 33], indirect discrimination [14, 21], redlining [7], statistical parity [12, 11, 22, 34], disparate mistreatment [32], or equality of opportunity [18]. However, all of these group benefit definitions invariably focus on achieving impact parity. We point interested readers to Feldman et al. [14] and Zafar et al. [32] regarding the discussion on this terminology.

Although not always explicitly sought, most of the above studies propose classifiers that also satisfy treatment parity in addition to impact parity, *i.e.*, they do not use the sensitive attribute $z$ in the decision making process. However, some of them [7, 18, 21] do not satisfy treatment parity since they resort to group-conditional classifiers, *i.e.*, $\boldsymbol{\theta} = \{\boldsymbol{\theta}_z\}_{z \in \mathcal{Z}}$. In such case, we can rewrite the above parity condition as:

$$\mathcal{B}_z(\boldsymbol{\theta}_z) = \mathcal{B}_{z'}(\boldsymbol{\theta}_{z'}) \quad \text{for all } z, z' \in \mathcal{Z}. \tag{2}$$

**Fairness beyond parity.** Given the above quality metrics, we can now formalize the two preference-based fairness notions introduced in Section 1.

— **Preferred treatment**: if a classifier $\boldsymbol{\theta}$ resorts to group-conditional classifiers, *i.e.*, $\boldsymbol{\theta} = \{\boldsymbol{\theta}_z\}_{z \in \mathcal{Z}}$, it is a preferred treatment classifier if each group sharing a sensitive attribute value $z$ benefits more from its corresponding group-conditional classifier $\boldsymbol{\theta}_z$ than it would benefit if it would be classified by any of the other group-conditional classifiers $\boldsymbol{\theta}_{z'}$, *i.e.*,

$$\mathcal{B}_z(\boldsymbol{\theta}_z) \geq \mathcal{B}_z(\boldsymbol{\theta}_{z'}) \quad \text{for all } z, z' \in \mathcal{Z}. \tag{3}$$

Note that, if a classifier $\boldsymbol{\theta}$ does not resort to group-conditional classifiers, *i.e.*, $\boldsymbol{\theta}_z = \boldsymbol{\theta}$ for all $z \in \mathcal{Z}$, it will be always be a preferred treatment classifier. If, in addition, such classifier ensures impact parity, it is easy to show that its utility cannot be larger than a preferred treatment classifier consisting of group-conditional classifiers.

— **Preferred impact**: a classifier $\boldsymbol{\theta}$ offers preferred impact over a classifier $\boldsymbol{\theta}'$ ensuring impact parity if it achieves higher group benefit for each sensitive attribute value group, *i.e.*,

$$\mathcal{B}_z(\boldsymbol{\theta}) \geq \mathcal{B}_z(\boldsymbol{\theta}') \quad \text{for all } z \in \mathcal{Z}. \tag{4}$$

One can also rewrite the above condition for group-conditional classifiers, *i.e.*, $\boldsymbol{\theta} = \{\boldsymbol{\theta}_z\}_{z \in \mathcal{Z}}$ and $\boldsymbol{\theta}' = \{\boldsymbol{\theta}'_z\}_{z \in \mathcal{Z}}$, as follows:

$$\mathcal{B}_z(\boldsymbol{\theta}_z) \geq \mathcal{B}_z(\boldsymbol{\theta}'_z) \quad \text{for all } z \in \mathcal{Z}. \tag{5}$$

Note that, given any classifier $\boldsymbol{\theta}'$ ensuring impact parity, it is easy to show that there will always exist a preferred impact classifier $\boldsymbol{\theta}$ with equal or higher utility.

**Connection to the fair division literature.** Our notion of preferred treatment is inspired by the concept of envy-freeness [5, 31] in the fair division literature. Intuitively, an envy-free resource division ensures that no user would *prefer the resources allocated* to another user over their own allocation. Similarly, our notion of preferred treatment ensures envy-free decision making at the level of sensitive attribute groups. Specifically, with preferred treatment classification, no sensitive attribute group would *prefer the outcomes from the classifier* of another group.

Our notion of preferred impact draws inspiration from the two-person bargaining problem [26] in the fair division literature. In a bargaining scenario, given a base resource allocation (also called the disagreement point), two parties try to divide some additional resources between themselves. If the parties cannot agree on a division, no party gets the additional resources, and both would only get the allocation specified by the disagreement point. Taking the resources to be the beneficial outcomes, and the disagreement point to be the allocation specified by the impact parity classifier, a preferred impact classifier offers enhanced benefits to all the sensitive attribute groups. Put differently, the group benefits provided by the preferred impact classifier Pareto-dominate the benefits provided by the impact parity classifier.

**On individual-level preferences.** Notice that preferred treatment and preferred impact notions are defined based on the group preferences, *i.e.*, whether a *group as a whole* prefers (or, gets more beneficial outcomes from) a given set of outcomes over another set. It is quite possible that a set of outcomes preferred by the group collectively is not preferred by certain *individuals* in the group. Consequently, one can extend our proposed notions to account for individual preferences as well, *i.e.*, a set of outcomes is preferred over another if *all* the individuals in the group prefer it. In the remainder of the paper, we focus on preferred treatment and preferred impact in the context of group preferences, and leave the case of individual preferences and its implications on the cost of achieving fairness for future work.

## 3 Training preferred classifiers

In this section, our goal is training preferred treatment and preferred impact group-conditional classifiers, *i.e.*, $\boldsymbol{\theta} = \{\boldsymbol{\theta}_z\}_{z \in \mathcal{Z}}$, that maximize utility given a training set $\mathcal{D} = \{(\boldsymbol{x}_i, y_i, z_i)\}_{i=1}^N$, where $(\boldsymbol{x}_i, y_i, z_i) \sim f(\boldsymbol{x}, y, z)$. In both cases, we will assume that:[2]

I. Each group-conditional classifier is a convex boundary-based classifier. For ease of exposition, in this section, we additionally assume these classifiers to be linear, *i.e.*, $\boldsymbol{\theta}_z(\boldsymbol{x}) = \boldsymbol{\theta}_z^T \boldsymbol{x}$, where $\boldsymbol{\theta}_z$ is a parameter that defines the decision boundary in the feature space. We relax the linearity assumption in Appendix A and extend our methodology to a non-linear SVM classifier.

II. The utility function $\mathcal{U}$ is defined as the overall accuracy of the group-conditional classifiers, *i.e.*,

$$\mathcal{U}(\boldsymbol{\theta}) = \mathbb{E}_{\boldsymbol{x},y}[\mathbb{I}\{\text{sign}(\boldsymbol{\theta}(\boldsymbol{x})) = y\}] = \sum_{z \in \mathcal{Z}} \mathbb{E}_{\boldsymbol{x},y|z}[\mathbb{I}\{\text{sign}(\boldsymbol{\theta}_z^T \boldsymbol{x}) = y\}]f(z). \qquad (6)$$

III. The group benefit $\mathcal{B}_z$ for users sharing the sensitive attribute value $z$ is defined as their average probability of being classified into the positive class, *i.e.*,

$$\mathcal{B}_z(\boldsymbol{\theta}) = \mathbb{E}_{\boldsymbol{x}|z}[\mathbb{I}\{\text{sign}(\boldsymbol{\theta}(\boldsymbol{x})) = 1\}] = \mathbb{E}_{\boldsymbol{x}|z}[\mathbb{I}\{\text{sign}(\boldsymbol{\theta}_z^T \boldsymbol{x}) = 1\}]. \qquad (7)$$

**Preferred impact classifiers.** Given a impact parity classifier with decision boundary parameters $\{\boldsymbol{\theta}_z'\}_{z \in \mathcal{Z}}$, one could think of finding the decision boundary parameters $\{\boldsymbol{\theta}_z\}_{z \in \mathcal{Z}}$ of a preferred impact classifier that maximizes utility by solving the following optimization problem:

$$
\begin{aligned}
&\underset{\{\boldsymbol{\theta}_z\}}{\text{minimize}} && -\tfrac{1}{N} \sum_{(\boldsymbol{x},y,z) \in \mathcal{D}} \mathbb{I}\{\text{sign}(\boldsymbol{\theta}_z^T \boldsymbol{x}) = y\} \\
&\text{subject to} && \sum_{\boldsymbol{x} \in \mathcal{D}_z} \mathbb{I}\{\text{sign}(\boldsymbol{\theta}_z^T \boldsymbol{x}) = 1\} \geq \sum_{\boldsymbol{x} \in \mathcal{D}_z} \mathbb{I}\{\text{sign}(\boldsymbol{\theta}_z'^T \boldsymbol{x}) = 1\} \quad \text{for all } z \in \mathcal{Z},
\end{aligned}
\qquad (8)
$$

where $\mathcal{D}_z = \{(\boldsymbol{x}_i, y_i, z_i) \in \mathcal{D} \mid z_i = z\}$ denotes the set of users in the training set sharing sensitive attribute value $z$, the objective uses an empirical estimate of the utility, defined by Eq. 6, and the preferred impact constraints, defined by Eq. 5, use empirical estimates of the group benefits, defined by Eq. 7. Here, note that the right hand side of the inequalities does not contain any variables and can be precomputed, *i.e.*, the impact parity classifiers $\{\boldsymbol{\theta}_z'\}_{z \in \mathcal{Z}}$ are given.

Unfortunately, it is very challenging to solve the above optimization problem since both the objective and constraints are nonconvex. To overcome this difficulty, we minimize instead a convex loss function $\ell_{\boldsymbol{\theta}}(\boldsymbol{x}, y)$, which is classifier dependent [6], and approximate the group benefits using a ramp (convex) function $r(x) = \max(0, x)$, *i.e.*,

$$\begin{aligned}
&\underset{\{\boldsymbol{\theta}_z\}}{\text{minimize}} && -\frac{1}{N} \sum_{(\boldsymbol{x}, y, z) \in \mathcal{D}} \ell_{\boldsymbol{\theta}_z}(\boldsymbol{x}, y) + \sum_{z \in \mathcal{Z}} \lambda_z \Omega(\boldsymbol{\theta}_z) \\
&\text{subject to} && \sum_{\boldsymbol{x} \in \mathcal{D}_z} \max(0, \boldsymbol{\theta}_z^T \boldsymbol{x}) \geq \sum_{\boldsymbol{x} \in \mathcal{D}_z} \max(0, \boldsymbol{\theta}_z'^T \boldsymbol{x}) \quad \text{for all } z \in \mathcal{Z},
\end{aligned} \tag{9}$$

which, for any convex regularizer $\Omega(\cdot)$, is a disciplined convex-concave program (DCCP) and thus can be efficiently solved using well-known heuristics [30]. For example, if we particularize the above formulation to group-conditional (standard) logistic regression classifiers $\boldsymbol{\theta}_z'$ and $\boldsymbol{\theta}_z$ and $L_2$-norm regularizer, then, Eq. 9 adopts the following form:

$$\begin{aligned}
&\underset{\{\boldsymbol{\theta}_z\}}{\text{minimize}} && -\frac{1}{N} \sum_{(\boldsymbol{x}, y, z) \in \mathcal{D}} \log p(y|\boldsymbol{x}, \boldsymbol{\theta}_z) + \sum_{z \in \mathcal{Z}} \lambda_z ||\boldsymbol{\theta}_z||^2 \\
&\text{subject to} && \sum_{\boldsymbol{x} \in \mathcal{D}_z} \max(0, \boldsymbol{\theta}_z^T \boldsymbol{x}) \geq \sum_{\boldsymbol{x} \in \mathcal{D}_z} \max(0, \boldsymbol{\theta}_z'^T \boldsymbol{x}) \quad \text{for all } z \in \mathcal{Z}.
\end{aligned} \tag{10}$$

where $p(y = 1|\boldsymbol{x}, \boldsymbol{\theta}_z) = \frac{1}{1 + e^{-\boldsymbol{\theta}_z^T \boldsymbol{x}}}$.

The constraints can similarly be added to other convex boundary-based classifiers like linear SVM. We further expand on particularizing the constraints for non-linear SVM in Appendix A.

**Preferred treatment classifiers.** Similarly as in the case of preferred impact classifiers, one could think of finding the decision boundary parameters $\{\boldsymbol{\theta}_z\}_{z \in \mathcal{Z}}$ of a preferred treatment classifier that maximizes utility by solving the following optimization problem:

$$\begin{aligned}
&\underset{\{\boldsymbol{\theta}_z\}}{\text{minimize}} && -\frac{1}{N} \sum_{(\boldsymbol{x}, y, z) \in \mathcal{D}} \mathbb{I}\{\text{sign}(\boldsymbol{\theta}_z^T \boldsymbol{x}) = y\} \\
&\text{subject to} && \sum_{\boldsymbol{x} \in \mathcal{D}_z} \mathbb{I}\{\text{sign}(\boldsymbol{\theta}_z^T \boldsymbol{x}) = 1\} \geq \sum_{\boldsymbol{x} \in \mathcal{D}_z} \mathbb{I}\{\text{sign}(\boldsymbol{\theta}_{z'}^T \boldsymbol{x}) = 1\} \quad \text{for all } z, z' \in \mathcal{Z},
\end{aligned} \tag{11}$$

where $\mathcal{D}_z = \{(\boldsymbol{x}_i, y_i, z_i) \in \mathcal{D} \,|\, z_i = z\}$ denotes the set of users in the training set sharing sensitive attribute value $z$, the objective uses an empirical estimate of the utility, defined by Eq. 6, and the preferred treatment constraints, defined by Eq. 3, use empirical estimates of the group benefits, defined by Eq. 7. Here, note that both the left and right hand side of the inequalities contain optimization variables.

However, the objective and constraints in the above problem are also nonconvex and thus we adopt a similar strategy as in the case of preferred impact classifiers. More specifically, we solve instead the following tractable problem:

$$\begin{aligned}
&\underset{\{\boldsymbol{\theta}_z\}}{\text{minimize}} && -\frac{1}{N} \sum_{(\boldsymbol{x}, y, z) \in \mathcal{D}} \ell_{\boldsymbol{\theta}_z}(\boldsymbol{x}, y) + \sum_{z \in \mathcal{Z}} \lambda_z \Omega(\boldsymbol{\theta}_z) \\
&\text{subject to} && \sum_{\boldsymbol{x} \in \mathcal{D}_z} \max(0, \boldsymbol{\theta}_z^T \boldsymbol{x}) \geq \sum_{\boldsymbol{x} \in \mathcal{D}_z} \max(0, \boldsymbol{\theta}_{z'}^T \boldsymbol{x}) \quad \text{for all } z, z' \in \mathcal{Z},
\end{aligned} \tag{12}$$

which, for any convex regularizer $\Omega(\cdot)$, is also a disciplined convex-concave program (DCCP) and can be efficiently solved.

## 4 Evaluation

In this section, we compare the performance of preferred treatment and preferred impact classifiers against unconstrained, treatment parity and impact parity classifiers on a variety of synthetic and real-world datasets. More specifically, we consider the following classifiers, which we train to maximize utility subject to the corresponding constraints:

— **Uncons**: an unconstrained classifier that resorts to group-conditional classifiers. It violates treatment parity—it trains a separate classifier per sensitive attribute value group—and potentially violates impact parity—it may lead to different benefits for different groups.

— **Parity**: a parity classifier that does not use the sensitive attribute group information in the decision making, but only during the training phase, and is constrained to satisfy both treatment parity— its decisions do not change based on the users' sensitive attribute value as it does not resort to group-conditional classifiers—and impact parity—it ensures that the benefits for all groups are the same. We train this classifier using the methodology proposed by Zafar et al. [33].

— **Preferred treatment**: a classifier that resorts to group-conditional classifiers and is constrained to satisfy preferred treatment—each group gets the highest benefit with its own classifier than any other group's classifier.

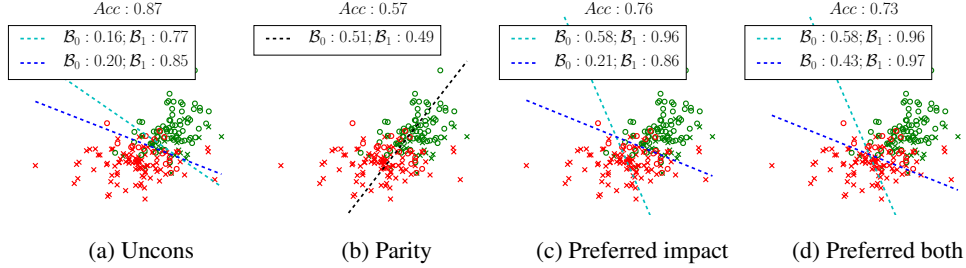

(a) Uncons      (b) Parity      (c) Preferred impact      (d) Preferred both

Figure 2: [Synthetic data] Crosses denote group-0 (points with $z = 0$) and circles denote group-1. Green points belong to the positive class in the training data whereas red points belong to the negative class. Each panel shows the accuracy of the decision making scenario along with group benefits ($\mathcal{B}_0$ and $\mathcal{B}_1$) provided by each of the classifiers involved. For group-conditional classifiers, cyan (blue) line denotes the decision boundary for the classifier of group-0 (group-1). Parity case (panel (b)) consists of just one classifier for both groups in order to meet the treatment parity criterion.

— **_Preferred impact_**: a classifier that resorts to group-conditional classifiers and is constrained to be preferred over the *Parity* classifier.

— **_Preferred both_**: a classifier that resort to group-conditional classifiers and is constrained to satisfy both *preferred treatment* and *preferred impact*.

For the experiments in this section, we use logistic regression classifiers with $L_2$-norm regularization. We randomly split the corresponding dataset into 70%-30% train-test folds 5 times, and report the average accuracy and group benefits in the test folds. Appendix B describes the details for selecting the optimal $L_2$-norm regularization parameters. Here, we compute utility ($\mathcal{U}$) as the overall accuracy of a classifier and group benefits ($\mathcal{B}_z$) as the fraction of users sharing sensitive attribute $z$ that are classified into the positive class. Moreover, the sensitive attribute is always binary, *i.e.*, $z \in \{0, 1\}$.

### 4.1 Experiments on synthetic data

**Experimental setup.** Following Zafar et al. [33], we generate a synthetic dataset in which the unconstrained classifier (*Uncons*) offers different benefits to each sensitive attribute group. In particular, we generate 20,000 binary class labels $y \in \{-1, 1\}$ uniformly at random along with their corresponding two-dimensional feature vectors sampled from the following Gaussian distributions: $p(\boldsymbol{x}|y = 1) = \mathcal{N}([2; 2], [5, 1; 1, 5])$ and $p(\boldsymbol{x}|y = -1) = \mathcal{N}([-2; -2], [10, 1; 1, 3])$. Then, we generate each sensitive attribute from the Bernoulli distribution $p(z = 1) = p(\boldsymbol{x}'|y = 1)/(p(\boldsymbol{x}'|y = 1) + p(\boldsymbol{x}'|y = -1))$, where $\boldsymbol{x}'$ is a rotated version of $\boldsymbol{x}$, *i.e.*, $\boldsymbol{x}' = [\cos(\pi/8), -\sin(\pi/8); \sin(\pi/8), \cos(\pi/8)]$. Finally, we train the five classifiers described above and compute their overall (test) accuracy and (test) group benefits.

**Results.** Figure 2 shows the trained classifiers, along with their overall accuracy and group benefits. We can make several interesting observations:

The **_Uncons_** classifier leads to an accuracy of $0.87$, however, the group-conditional boundaries and high disparity in treatment for the two groups ($0.16$ vs. $0.85$) mean that it satisfies neither treatment parity nor impact parity. Moreover, it leads to only a small violation of preferred treatment—benefits for group-0 would increase slightly from $0.16$ to $0.20$ by adopting the classifier of group-1. However, this will not always be the case, as we will later show in the experiments on real data.

The **_Parity_** classifier satisfies both treatment and impact parity, however, it does so at a large cost in terms of accuracy, which drops from $0.87$ for *Uncons* to $0.57$ for *Parity*.

The **_Preferred treatment_** classifier (not shown in the figure), leads to a minor change in decision boundaries as compared to the *Uncons* classifier to achieve preferred treatment. Benefits for group-0 (group-1) with its own classifier are $0.20$ $(0.84)$ as compared to $0.17$ $(0.83)$ while using the classifier of group-1 (group-0). The accuracy of this classifier is $0.87$.

The **_Preferred impact_** classifier, by making use of a looser notion of fairness compared to impact parity, provides higher benefits for both groups at a much smaller cost in terms of accuracy than the *Parity* classifier ($0.76$ vs. $0.57$). Note that, while the *Parity* classifier achieved equality in benefits by misclassifying *negative examples from group-0* into the positive class and misclassifying *positive*

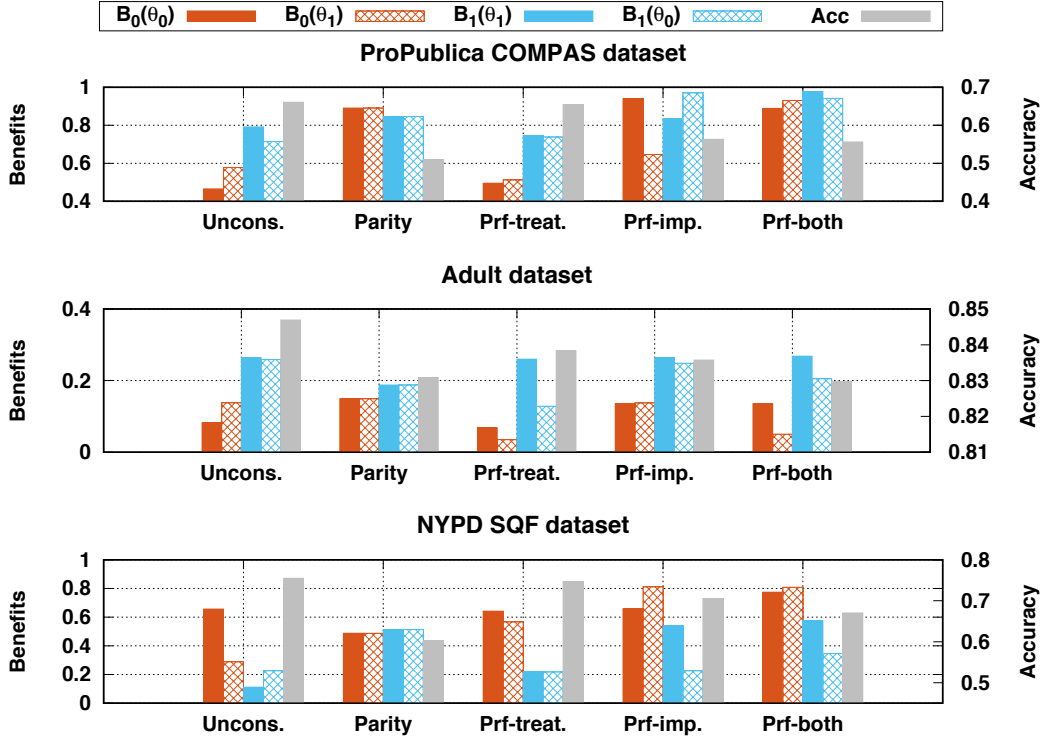

Figure 3: The figure shows the accuracy and benefits received by the two groups for various decision making scenarios. 'Prf-treat.', 'Prf-imp.', and 'Prf-both' respectively correspond to the classifiers satisfying preferred treatment, preferred impact, and both preferred treatment and impact criteria. Sensitive attribute values 0 and 1 denote blacks and whites in ProPublica COMPAS dataset and NYPD SQF datasets, and women and men in the Adult dataset. $\mathcal{B}_i(\boldsymbol{\theta}_j)$ denotes the benefits obtained by group $i$ when using the classifier of group $j$. For the *Parity* case, we train just one classifier for both the groups, so the benefits do not change by adopting other group's classifier.

*examples from group-1* into the negative class, the *Preferred impact* classifier only incurs the former type of misclassifications. However, the outcomes of the *Preferred impact* classifier do not satisfy the preferred treatment criterion: group-1 would attain higher benefit if it used the classifier of group-0 (0.96 as compared to 0.86).

Finally, the classifier that satisfies preferred treatment and preferred impact (***Preferred both***) achieves an accuracy and benefits at par with the *Preferred impact* classifier.

We present the results of applying our fairness constraints on a non linearly-separable dataset with a SVM classifier with a radial basis function (RBF) kernel in Appendix C.

### 4.2 Experiments on real data

**Dataset description and experimental setup.** We experiment with three real-world datasets: the COMPAS recidivism prediction dataset compiled by ProPublica [23], the Adult income dataset from UCI machine learning repository [2], and the New York Police Department (NYPD) Stop-question-and-frisk (SQF) dataset made publicly available by NYPD [1]. These datasets have been used by a number of prior studies in the fairness-aware machine learning literature [14, 29, 32, 34, 33].

In the COMPAS dataset, the classification task is to predict whether a criminal defendant would recidivate within two years (negative class) or not (positive class); in the Adult dataset, the task is to predict whether a person earns more than 50K USD per year (positive class) or not; and, in the SQF dataset, the task is to predict whether a pedestrian should be stopped on the suspicion of having an illegal weapon or not (positive class). In all datasets, we assume being classified as positive to be the beneficial outcome. Additionally, we divide the subjects in each dataset into two sensitive attribute value groups: women (group-0) and men (group-1) in the Adult dataset and blacks (group-0) and whites (group-1) in the COMPAS and SQF datasets. The supplementary material

(Appendix D) contains more information on the sensitive and the non-sensitive features as well as the class distributions.[3]

**Results.** Figure 3 shows the accuracy achieved by the five classifiers described above along with the benefits they provide for the three datasets. We can draw several interesting observations:[4]

In all cases, the ***Uncons*** classifier, in addition to violating treatment parity (a separate classifier for each group) and impact parity (high disparity in group benefits), also violates the preferred treatment criterion (in all cases, at least one of group-0 or group-1 would benefit more by adopting the other group's classifier). On the other hand, the ***Parity*** classifier satisfies the treatment parity and impact parity but it does so at a large cost in terms of accuracy.

The ***Preferred treatment*** classifier provides a much higher accuracy than the *Parity* classifier—its accuracy is at par with that of the *Uncons* classifier—while satisfying the preferred treatment criterion. However, it does not meet the preferred impact criterion. The ***Preferred impact*** classifier meets the preferred impact criterion but does not always satisfy preferred treatment. Moreover, it also leads to a better accuracy then *Parity* classifier in all cases. However, the gain in accuracy is more substantial for the SQF datasets as compared to the COMPAS and Adult dataset.

The classifier satisfying preferred treatment and preferred impact (***Preferred both***) has a somewhat underwhelming performance in terms of accuracy for the Adult dataset. While the performance of this classifier is better than the *Parity* classifier in the COMPAS dataset and NYPD SQF dataset, it is slightly worse for the Adult dataset.

In summary, the above results show that ensuring either preferred treatment or preferred impact is less costly in terms of accuracy loss than ensuring parity-based fairness, however, ensuring both preferred treatment and preferred impact can lead to comparatively larger accuracy loss in certain datasets. We hypothesize that this loss in accuracy may be partly due to splitting the number of available samples into groups during training—each group-conditional classifier use only samples from the corresponding sensitive attribute group—hence decreasing the effectiveness of empirical risk minimization.

# 5   Conclusion

In this paper, we introduced two preference-based notions of fairness—preferred treatment and preferred impact—establishing a previously unexplored connection between fairness-aware machine learning and the economics and game theoretic concepts of envy-freeness and bargaining. Then, we proposed tractable proxies to design boundary-based classifiers satisfying these fairness notions and experimented with a variety of synthetic and real-world datasets, showing that preference-based fairness often allows for greater decision accuracy than existing parity-based fairness notions.

Our work opens many promising venues for future work. For example, our methodology is limited to convex boundary-based classifiers. A natural follow up would be to extend our methodology to other types of classifiers, *e.g.*, neural networks and decision trees. In this work, we defined preferred treatment and preferred impact in the context of group preferences, however, it would be worth revisiting the proposed definitions in the context of individual preferences. The fair division literature establishes a variety of fairness axioms [26] such as Pareto-optimality and scale invariance. It would be interesting to study such axioms in the context of fairness-aware machine learning.

Finally, we note that while moving from parity to preference-based fairness offers many attractive properties, we acknowledge it may not always be the most appropriate notion, *e.g.*, in some scenarios, parity-based fairness may very well present the eventual goal and be more desirable [3].

**Acknowledgments**

AW acknowledges support by the Alan Turing Institute under EPSRC grant EP/N510129/1, and by the Leverhulme Trust via the CFI.

## Footnotes

[1] Although it is quite possible that certain *individuals* from the group may not prefer the set that maximizes the benefit for the *group as a whole*.

[2] Exploring the relaxations of these assumptions is a very interesting avenue for future work.

[3]Since the SQF dataset is highly skewed in terms of class distribution ($\sim 97\%$ points in the positive class) resulting in a trained classifier predicting all points as positive (yet having $97\%$ accuracy), we subsample the dataset to have equal class distribution. Another option would be using penalties proportional to the size of the class, but we observe that an unconstrained classifier with class penalties gives similar predictions as compared to a balanced dataset. We decided to experiment with the balanced dataset since the accuracy drops in this dataset are easier to interpret.

[4]The unfairness in the SQF dataset is different from what one would expect [27]—an unconstrained classifier gives more benefits to blacks as compared to whites. This is due to the fact that a larger fraction of stopped whites were found to be in possession on an illegal weapon (Tables 3 and 4 in Appendix D).

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
