[Supplementary Material]

# A  Particularizing fairness constraints for non-linear SVM

For a non-linear SVM, given a training dataset $\mathcal{D} = \{(\boldsymbol{x}_i, y_i, z_i)\}_{i=1}^{N}$, one typically finds the optimal decision boundary parameters $\boldsymbol{\alpha}$ by solving the dual of the corresponding optimization problem [16], which takes the following form:

$$
\begin{aligned}
\underset{\boldsymbol{\alpha}}{\text{minimize}} \quad & \tfrac{1}{2}\boldsymbol{\alpha}^T \mathbf{G} \boldsymbol{\alpha} - \mathbf{1}^T \boldsymbol{\alpha} \\
\text{subject to} \quad & 0 \leq \boldsymbol{\alpha} \leq C, \\
& \boldsymbol{y}^T \boldsymbol{\alpha} = 0
\end{aligned}
$$

where $\boldsymbol{\alpha} = [\alpha_1, \alpha_2, \dots, \alpha_N]^T$ are the optimization variables specifying the decision boundary, $\boldsymbol{y} = [y_1, y_2, \dots, y_N]^T$ are the class labels, and $\boldsymbol{G}$ is the $N \times N$ Gram matrix with $\boldsymbol{G}_{i,j} = y_i y_j k(\boldsymbol{x}_i, \boldsymbol{x}_j)$. Here, the kernel function, $k(\boldsymbol{x}_i, \boldsymbol{x}_j) = \phi(\boldsymbol{x}_i) \cdot \phi(\boldsymbol{x}_j)$ denotes the inner product between a pair of transformed feature vectors. Then, given an unknown data point $\boldsymbol{x}$, one computes $\hat{y} = \text{sign}(\boldsymbol{\alpha}(\boldsymbol{x}))$ where $\boldsymbol{\alpha}(\boldsymbol{x}) = \sum_{i=1}^{N} \alpha_i y_i k(\boldsymbol{x}, \boldsymbol{x}_i)$ where $\boldsymbol{\alpha}(\boldsymbol{x})$ can still be interpreted as the signed distance from the decision boundary.

Given, this specification, one can particularize Eq. 9 for training group-conditional preferred impact non-linear SVMs as:

$$
\begin{aligned}
\underset{\{\boldsymbol{\alpha}_z\}}{\text{minimize}} \quad & \sum_{z \in \mathcal{Z}} \tfrac{1}{2}\boldsymbol{\alpha}_z^T \mathbf{G}_z \boldsymbol{\alpha}_z - \mathbf{1}^T \boldsymbol{\alpha}_z \\
\text{subject to} \quad & 0 \leq \boldsymbol{\alpha}_z \leq C_z \quad \text{for all } z \in \mathcal{Z}, \\
& \boldsymbol{y}_z^T \boldsymbol{\alpha}_z = 0 \quad \text{for all } z \in \mathcal{Z}, \\
& \sum_{\boldsymbol{x} \in \mathcal{D}_z} \max(0, \boldsymbol{\alpha}_z(\boldsymbol{x})) \geq \sum_{\boldsymbol{x} \in \mathcal{D}_z} \max(0, \boldsymbol{\alpha}_z^{'}(\boldsymbol{x})) \quad \text{for all } z \in \mathcal{Z},
\end{aligned}
$$

where $\{\boldsymbol{\alpha}_z^{'}\}_{z \in \mathcal{Z}}$ are the given parity impact classifiers and $\mathbf{G}_z$ and $\boldsymbol{y}_z$ denote the Gram matrix and class label vector for the sensitive attribute group $z$.

One can similarly particularize Eq. 12 for training group-conditional preferred treatment non-linear SVMs as:

$$
\begin{aligned}
\underset{\{\boldsymbol{\alpha}_z\}}{\text{minimize}} \quad & \sum_{z \in \mathcal{Z}} \tfrac{1}{2}\boldsymbol{\alpha}_z^T \mathbf{G}_z \boldsymbol{\alpha}_z - \mathbf{1}^T \boldsymbol{\alpha}_z \\
\text{subject to} \quad & 0 \leq \boldsymbol{\alpha}_z \leq C_z \quad \text{for all } z \in \mathcal{Z}, \\
& \boldsymbol{y}_z^T \boldsymbol{\alpha}_z = 0 \quad \text{for all } z \in \mathcal{Z}, \\
& \sum_{\boldsymbol{x} \in \mathcal{D}_z} \max(0, \boldsymbol{\alpha}_z(\boldsymbol{x})) \geq \sum_{\boldsymbol{x} \in \mathcal{D}_z} \max(0, \boldsymbol{\alpha}_{z'}(\boldsymbol{x})) \quad \text{for all } z, z' \in \mathcal{Z}.
\end{aligned}
$$

One can similarly add the constraints to the non-linear SVM in the primal form [8].

# B  Experimental details

In this section, we provide details for selecting the optimal $L_2$-norm regularization parameters ($\lambda$) for the experiments performed in Section 4. For performing the validation procedure below, we first split the training dataset ($\mathcal{D}_{train}$) further into a 70%-30% train set ($\mathcal{D}_{tr}$) and a validation set ($\mathcal{D}_{val}$). Then, for a given range $L = \{\lambda_1, \lambda_2, \dots, \lambda_k\}$ of candidate values, we select the optimal ones as follows.

**Unconstrained and parity classifiers.** These cases consist of *training one classifier at a time*. For the unconstrained classifier, we train one classifier for each sensitive attribute group separately. For the parity classifier, we train one classifier for all groups.

For each value of $\lambda \in L$, we train the classifier on $\mathcal{D}_{tr}$, and choose the one that provides best accuracy on the validation set $\mathcal{D}_{val}$. We call it $\lambda^{opt}$. We then train the classifier on the whole training dataset $\mathcal{D}_{train}$ with $\lambda^{opt}$.

**Preferentially fair classifiers.** Training preferentially fair classifiers in Eq. 9 and Eq. 12 consists of jointly minimizing the objective function for both groups while satisfying the fairness constraints. For training these classifiers for two groups (say group-0 and group-1), we take all combinations of $\lambda_0, \lambda_1 \in L$, and choose the combination that provides best accuracy on $\mathcal{D}_{val}$ while satisfying the constraints. For real-world datasets, we specify the following tolerance level for the constraints: for a given pair of $\lambda_0, \lambda_1 \in L$, we consider the constraints to be satisfied if the observed value of group benefits $\mathcal{B}_z$ in the validation set $\mathcal{D}_{val}$ and the desired value are at least within 90% of each other, and additionally, the difference between them is no more that $0.03$. We notice that setting hard thresholds

|     |     |     |     |
| :-: | :-: | :-: | :-: |
| $Acc: 0.96; \mathcal{B}_0: 0.07; \mathcal{B}_1: 0.84$ | $Acc: 0.61; \mathcal{B}_0: 0.36; \mathcal{B}_1: 0.38$ | $Acc: 0.93; \mathcal{B}_0: 0.15; \mathcal{B}_1: 0.83$ | $Acc: 0.84; \mathcal{B}_0: 0.36; \mathcal{B}_1: 0.88$ |
| $Acc: 0.96; \mathcal{B}_0: 0.17; \mathcal{B}_1: 0.87$ | $Acc: 0.61; \mathcal{B}_0: 0.36; \mathcal{B}_1: 0.38$ | $Acc: 0.93; \mathcal{B}_0: 0.16; \mathcal{B}_1: 0.86$ | $Acc: 0.84; \mathcal{B}_0: 0.18; \mathcal{B}_1: 0.87$ |
| (a) Uncons | (b) Parity | (c) Preferred treatment | (d) Preferred impact |

Figure 4: [Non linearly separable synthetic data] Crosses denote group-0 (points with $z = 0$) and circles denote group-1. Green points belong to the positive class in the training data whereas red points belong to the negative class. Each panel shows the classifiers with top row containing the classifiers for group-0 and the bottom for group-1, along with the overall accuracy as well as the group benefits ($\mathcal{B}_0$ and $\mathcal{B}_1$) provided by each of the classifiers involved. For parity classifier, no group-conditional classifiers are allowed, so both top and bottom row contain the same classifier.

with no tolerance on real-world datasets sometimes leads to divergent solutions in terms of group benefits. We hypothesize that this effect may be due to the underlying variance between $D_{tr}$ and $D_{val}$.

## C  Experiments with non-linear SVM

In this section, we demonstrate the effectiveness of our constraints in ensuring fairness on a non linearly-separable dataset with a SVM classifier using radial basis function (RBF) kernel.

Following the setup of Zafar et al. [33], we generated a synthetic dataset consisting of $4,000$ user binary class labels uniformly at random. We then assign a 2-dimensional user feature vector to each label by drawing samples from the following distributions:

$$p(\boldsymbol{x}|y = 1, \beta) = \beta N([2; 2], [5\ 1; 1\ 5]) + (1 - \beta)N([-2; -2], [10\ 1; 1\ 3])$$
$$p(\boldsymbol{x}|y = -1, \beta) = \beta N([4; -4], [4\ 4; 2\ 5]) + (1 - \beta)N([-4; 6], [6\ 2; 2\ 3])$$

where $\beta \in \{0, 1\}$ is sampled from Bernoulli(0.5). We then generate the corresponding user sensitive attributes $z$ by applying the same rotation as detailed in Section 4.

We then train the various classifiers described in Section 4. The results are shown in Figure 4. Top row in the figure shows the group-conditional classifiers for group-0, whereas, the bottom row shows the ones for group-1. For the case of parity classifier, due to treatment parity condition, both groups use the same classifier.

The *Uncons* classifier leads to an accuracy of $0.96$, however, the group-conditional classifiers lead to high disparity in beneficial outcomes for both groups ($0.07$ vs. $0.87$). The classifier also leads to a violation of preferred treatment—the benefits for group-0 would increase from $0.07$ with its own classifier to $0.17$ with the classifier of group-1.

The *Parity* classifier satisfies both treatment and impact parity, however, it does so at a large cost in terms of accuracy, which drops from $0.96$ for *Uncons* to $0.61$ for *Parity*.

The *Preferred treatment* classifier, adjusts the decision boundary for group-0 to remove envy and does so at a small cost in accuracy (from $0.96$ to $0.93$).

The *Preferred impact* classifier, by making use of the relaxed parity-fairness conditions, provides higher or equal benefits for both groups at a much smaller cost in terms of accuracy than the *Parity* classifier ($0.84$ vs. $0.61$). The preferred impact classifier in this case also satisfies the preferred treatment criterion.

# D  Dataset statistics

For the ProPublica COMPAS dataset, we use the same non-sensitive features as used by Zafar et al. [32]. The non-sensitive features include number of prior offenses, the degree of the arrest charge (misdemeanor or felony), *etc*. The class and sensitive attribute distribution in the dataset is in Table 1.

Table 1: Recidivism rates in ProPublica COMPAS data for both races.

| Race | Yes (-ve) | No (+ve) | Total |
|------|-----------|----------|-------|
| Black | $1,661(52\%)$ | $1,514(48\%)$ | $3,175$ |
| White | $8,22(39\%)$ | $1,281(61\%)$ | $2,103$ |
| Total | $2,483(47\%)$ | $2,795(53\%)$ | $5,278$ |

For Adult dataset [2], we use the same non-sensitive features as a number of prior studies [14, 33, 34] on fairness-aware learning. The non-sensitive features include educational level of the person, number of working hours per week, etc. The class and sensitive attribute distribution in the dataset is as follows in Table 2.

Table 2: High income ($\geq$ 50K USD) in Adult data for both genders.

| Gender | Yes (+ve) | No (-ve) | Total |
|--------|-----------|----------|-------|
| Males | 9,539(31%) | 20,988(69%) | 30,527 |
| Females | 1,669(11%) | 13,026(89%) | 14,695 |
| Total | 34,014(75%) | 11,208(25%) | 45,222 |

For the NYPD SQF dataset [1], we use the same prediction task and non-sensitive features as used by Goel et al. [29]. We only use the stops made in 2012. The prediction task is, whether a pedestrian stopped on the suspicion of having a weapon actually possesses a weapon or not. The non-sensitive features include proximity to a crime scene, age/build of a person, and so on. Finally, as explained in Section 4, since the original dataset (Table 3) is highly skewed towards the positive class we subsample the majority class (positive) to match the size of the minority (negative) class.

Table 3: Persons found to be in possession of a weapon in 2012 NYPD SQF dataset (original).

| Race | Yes (-ve) | No (+ve) | Total |
|------|-----------|----------|-------|
| Black | $2,113(3\%)$ | $77,337(97\%)$ | $79,450$ |
| White | $803(15\%)$ | $4,616(85\%)$ | $5,419$ |
| Total | $2,916(3\%)$ | $81,953(97\%)$ | $84,869$ |

Table 4: Persons found to be in possession of a weapon in 2012 NYPD SQF dataset (class-balanced).

| Race | Yes (-ve) | No (+ve) | Total |
|------|-----------|----------|-------|
| Black | $2,113(43\%)$ | $2,756(57\%)$ | $4,869$ |
| White | $803(83\%)$ | $160(17\%)$ | $963$ |
| Total | $2,916(50\%)$ | $2,916(50\%)$ | $5,832$ |