[Reviews · NeurIPS 2017]

Reviewer 1



This paper introduces a new notion of fairness in decision-making. While there are already many notions of fairness proposed in past work, I found the proposed definition unique in the sense that it takes an game-theoretic approach to fair decisions. Specifically, the authors propose an envy-free criterion: once the decisions are made, no group of people with a protected attribute would prefer to be (counterfactually) in another group. Both demographic parity and equality of opportunity metrics can be weakened to yield their envy-free equivalents. Once defined, the authors define a straightforward way to design a learning algorithm with constraints. The results show that envy-free leads to a lower reduction in accuracy than other stricter forms of fairness. Overall, this is a good conceptual addition to the discussion on fairness, making the connections to past work in game theory and economics.

Reviewer 2



ML models control many aspects of our life such as our bank decision on approving a loan or a medical centers that use ML to make treatment decision. Therefore, it is paramount that these models are fair, even in cases where the data contains biases. This study follows previous studies in the field but argues that the definition of fairness provided in previous studies might pose a too high bar which results in unnecessary cost in terms of accuracy. Instead they present an alternative fairness definition, presenting an algorithm to apply it for linear classifiers together with some empirical results. I find this paper very interesting and relevant but lacking in several aspects. I think that it will benefit from some additional depth which will make it a significant contribution. Here are some issues that I find with this paper: 1. I am not sure that the treatment of the bias is comprehensive enough. For example, if z is a gender feature. A classifier may ignore this feature altogether but still introduce bias by inferring gender from other features such that height, weight, facial image, or even the existence of male/female reproducing organs. I could not figure out from the text how this issue is treated. This is important that in the current definition, if a model never treats a person with male reproducing organs for cancer, this model may be gender-fair. The paragraph at lines 94-100 gives a short discussion that relates to this issue but it should be made more explicit and clear. 2. There is a sense of over-selling in this paper. The abstract suggests that there is an algorithmic solution but only in section 3 an assumption is added that the classification model is linear. This is a non-trivial constraint but there is no discussion on scenarios in which linear models are to be used. Some additional comments that the authors may wish to consider: 3. The introduction can benefit from concrete examples. For example, you can take the case of treating cancer: while male and female should receive equally good treatment, it does not mean that female should be treated for prostate cancer or male should be treated for breast cancer in the same rates. 4. Lines 66-74 are repetition of previous paragraphs 5. Line 79: “may” should be “many” 6. When making the approximations (lines 171-175) you should show that these are good approximation in a sense that a solution to the approximated problem will be a good solution to the original problem. It may be easier to state the original problem as (9) and avoid the approximation discussion.

Reviewer 3



This work addresses fairness in machine learning by adopting notions like envy freeness from the fair division literature. The basic idea is to create classifiers such that no group prefers ("envies") its treatment under a classifer to some other group, under some sane conditions to ensure the classifier is meaningful. The paper looks at fairness in two contexts, treatment (roughly speaking, how the decision is made) and impact (roughly speaking, the impact of a decision w.r.t. a specific group). As with any economic problem, there will be a tradeoff between efficiency and fairness; in the context of this paper, that is the tradeoff between the "utility" achieved by the classifier and the "group benefit" received by the different groups individually. The paper directly translates notions like E-F into this language, discusses ways to train classifiers with added E-F constraints, and provides some nice experimental results exploring what unfair, parity-based fair, and their proposed social choice-style fair classifiers do. In terms of "core CS," the contribution here is not huge. Indeed, the paper takes a concept from an orthogonal literature (computational social choice) and applies it quite directly to general supervised learning as a hard constraint, and then discusses a reasonable tweak to help with the non-convexness of that new problem. I absolutely don't see this as a problem, though; indeed, the paper is well-motivated, reads quite nicely, and presents a simple adaptation of a very grounded (from the social choice side) concept to fairness in ML, which is still a pretty brittle and poorly-understood area of ML. While I don't think group-based preference methods like E-F would be appropriate for all "fairness in ML" style applications, I find the method compelling for many situations.